# Psychosocial Distress in Parents with Children Awaiting Surgery during the COVID-19 Pandemic

**DOI:** 10.3390/children9010087

**Published:** 2022-01-09

**Authors:** David Forner, Patricia K. Leslie, Abdullah Aldaihani, Michael Bezuhly, Christopher W. Noel, David Horne, Simon Walling, Johane Robitaille, Dawn L. MacLellan, Ron El-Hawary, Karl Logan, Rodrigo Romao, Robert LaRoche, Suvro Sett, Robin Urquhart, Paul Hong

**Affiliations:** 1Division of Otolaryngology—Head & Neck Surgery, Dalhousie University, Halifax, NS B3H 4R2, Canada; pleslie0@gmail.com (P.K.L.); Aldaihani@dal.ca (A.A.); paul.hong@iwk.nshealth.ca (P.H.); 2Institute of Health Policy, Management and Evaluation, University of Toronto, Toronto, ON M5S 1A1, Canada; christopher.noel@mail.utoronto.ca; 3Division of Plastic Surgery, Dalhousie University, Halifax, NS B3H 4R2, Canada; michael.bezuhly@iwk.nshealth.ca; 4Department of Otolaryngology—Head & Neck Surgery, University of Toronto, Toronto, ON M5S 1A1, Canada; 5Division of Cardiac Surgery, Dalhousie University, Halifax, NS B3H 4R2, Canada; david.horne@iwk.nshealth.ca (D.H.); Suvro.sett@iwk.nshealth.ca (S.S.); 6Department of Neurosurgery, Dalhousie University, Halifax, NS B3H 4R2, Canada; simon.walling@iwk.nshealth.ca; 7Department of Ophthalmology and Visual Sciences, Dalhousie University, Halifax, NS B3H 4R2, Canada; jrobitai@dal.ca (J.R.); robert.laroche@dal.ca (R.L.); 8Department of Urology, Dalhousie University, Halifax, NS B3H 4R2, Canada; dawn.maclellan@iwk.nshealth.ca; 9Division of Orthopedic Surgery, Dalhousie University, Halifax, NS B3H 4R2, Canada; ron.el-hawary@iwk.nshealth.ca (R.E.-H.); karl.logan@iwk.nshealth.ca (K.L.); 10Division of Pediatric General and Thoracic Surgery, Dalhousie University, Halifax, NS B3H 4R2, Canada; rodrigo.romao@iwk.nshealth.ca; 11Department of Surgery, Dalhousie University, Halifax, NS B3H 4R2, Canada; robin.urquhart@nshealth.ca; 12Department of Community Health and Epidemiology, Dalhousie University, Halifax, NS B3H 4R2, Canada

**Keywords:** COVID-19, psychosocial distress, waiting lists, pediatric surgery

## Abstract

Due to resource restrictions related to the COVID-19 pandemic, many pediatric patients are facing substantial delays for surgery, potentially resulting in additional distress for caregivers. We aimed to assess the experiences and psychosocial distress of parents during COVID-19 as they relate to the pandemic, waiting for surgery, and the combined effects of both events. The was a cross-sectional qualitative study. Parents with children who faced treatment delays during the initial wave of the COVID-19 pandemic for elective, non-emergent procedures across a variety of surgical specialties were recruited. Semi-structured telephone interviews and thematic analysis were utilized. Thematic saturation was reached with eighteen participants. Four themes were identified: coping with COVID-19, distress levels, quality and nature of communication with the surgical team, and the experience of COVID-19 related hospital restrictions. Participants reported varying levels of distress due to the delay in surgery, such as the fear of developmental delay or disease progression for their child. They also indicated their own physical and mental health had been impacted by emotional distress related to both COVID-19 and delays in treatment. Most participants experienced the COVID-19-related hospital restrictions as distressing. This related predominantly to limiting in-hospital caregivers to only one caregiver. Participants were found to have substantial levels of psychosocial distress. Targeted social and emotional support may be helpful in reducing parental distress as the pandemic timeframe continues. Within the limits of individual health systems, reducing restrictions to the number of allowed care givers may help allay distress felt by parents.

## 1. Introduction

Awaiting surgery can be distressing for patients and parents alike [1]. Anxiety and other negative emotions experienced by both the patient and their caregivers may result in compounded feelings of distress [2]. Unfortunately, despite extensive efforts to identify and reduce surgical wait times [3], parents often believe the wait time experienced by their child is too long [1]. During the COVID-19 pandemic, restrictions on operating room availability has been necessary at most institutions. Hundreds of thousands of surgical cases have been canceled in Canada alone [4]. Patients awaiting routine, non-emergent surgery are facing substantial treatment delays beyond those typically experienced.

The COVID-19 pandemic has also caused disruption and lifestyle changes for many families. Child and adolescent mental health is substantially worse during the pandemic, with nearly 40% of children experiencing anxiety- and depression-like symptoms in one study [5]. Parental stress has also risen compared to the pre-pandemic period [6]. The requirement to plan new financial, employment, care giving, and child educational routines have served to escalate parental stress [7]. Typical stress coping strategies, such as socialization, may be limited during periods of public health restrictions, further exacerbating this dilemma.

Given the association of COVID-19 on both surgical wait times and parental distress, this study sought to explore the psychosocial outcomes of parents with children facing prolonged surgical wait times during the COVID-19 pandemic.

## 2. Methods

### 2.1. Study Design

This is a cross-sectional qualitative study of parents with children who faced treatment delays at an academic, tertiary care pediatric hospital in Halifax, Nova Scotia, Canada within a universal, single-payer healthcare system. Institutional Review Board approval was obtained from the IWK Health Centre (IWK REB# 1025774, Halifax, NS, Canada) on 27 May 2020. This study is reported in adherence to the Standards for Reporting Qualitative Research (SRQR) guidelines [8].

### 2.2. Population

The population of interest was caregivers of children awaiting elective, non-emergent surgery during the first wave of the COVID-19 pandemic, beginning in March 2020. Ultimately, all caregivers recruited to this study identified themselves as parents of the child. Procedures included adenotonsillectomy, myringotomy, and tympanostomy tube placement, cleft palate repair, strabismus correction surgery, scoliosis correction surgery, herniorrhaphy, atrial septal defect repair, orchidopexy, and femoroacetabular impingement repair.

Purposive sampling was used to select participants representative of the population of interest [9]. Representative criteria included age, sex, and the procedure the patient was awaiting. Parents were recruited from 17 June to 28 July 2020. Participants were recruited until theory saturation was reached [10], defined as no new unique domains or themes generated across interviews. Both purposive sampling and theory saturation are unique to qualitative research. In quantitative research, where statistical analysis and the testing of pre-determined hypotheses necessitates statistical power, large sample sizes are encouraged. In qualitative analysis, theory saturation may be reached with fewer participants. As such, in qualitative research, where statistical power is not required for hypothesis testing, large sample sizes are not required and do not contribute to further new data development. Potential participants were approached by an individual within their child’s circle of care, and subsequently by a member of the research team (PKL). Participants underwent an informed consent process and were provided with an honorarium gift card.

### 2.3. Data Collection

Participants underwent semi-structured telephone interviews with a member of the research team (PKL). Interviews were conducted with the aid of an interview guide (Appendix A). The semi-structured interview guide sought to elicit participant experiences with the pandemic, surgical delays, and their impact on the participant and their children. Open-ended questions were utilized. For example, we asked, “Tell me about your experiences since your child’s surgery was delayed because of the COVID-19 pandemic.” Interview recordings were transcribed verbatim.

### 2.4. Qualitative Analysis

This study was carried out through an interpretivism research paradigm whereby participant experiences were interpreted in the context of their environment and lived experiences [11]. Reflexive thematic analysis, herein referred to as thematic analysis, under the Braun and Clarke framework [12] was used for analysis. Under this framework, thematic analysis is completed via six phases, including familiarization, generating initial codes, searching for themes, reviewing themes, defining and naming themes, and reporting the data. In this study, we aimed to utilize thematic analysis to describe the lived experiences of a particular group. To facilitate this, interview data were coded, and themes were conceptualized and progressively refined. In order to promote credibility, and dependability of our analysis [13,14], the first three interviews were independently coded (P.K.L., R.U.) and used to develop a novel codebook using inductive coding performed in an iterative fashion. The remaining interviews were coded to develop domains (P.K.L., R.U.), as a means to bridge conceptualization of themes in the next phase. Themes summarizing these domains were thus developed (P.K.L., R.U.). Consensus discussion, elaboration, and review of themes was performed with multiple team members (D.F., P.K.L., R.U., P.H.) to support reliability in our procedures [13]. Supporting quotations were extracted for each theme and its underlying domains (PKL, RU). Data handling was performed with NVivo 12 (QSR International). Reflexivity statements are found in Appendix A.

## 3. Results

### 3.1. Participants

The total number of eligible participants approached was 30. Ten participants did not respond to the research assistant’s contact for interview, while two participants no longer wished to take part in the interview. Therefore, once theory saturation had been reached, eighteen participants were ultimately included in the study at the completion of the recruitment process. The mean age of participants was 34.2 years (SD: 4.4, range 28–43). The mean age of the children was 65.5 months (SD: 55.5, range 7–192). Participant and child characteristics are found in Table 1. All parent participants were female (100%), and the majority of children were male (72%). Most parents had college/undergraduate education or higher (73%) and were married (67%). The mean interview length was 14.9 min (range 5–37 min)

### 3.2. Qualitative Analysis

Four themes were identified from the dataset: the nature and quality of communication with their child’s surgical team (communication), parents’ levels of frustration and distress due to their child’s delay in surgery (levels of distress), families’ coping with surgical delay during the COVID-19 pandemic (coping with COVID-19), and the experience of COVID-19 related hospital restrictions (hospital restrictions). Supporting codes and domains are found in Appendix A.

### 3.3. Communication

Participants discussed varied experiences in terms of the nature and quality of communication with the surgical team during the delay. Most participants were informed about the COVID-19-related delay via telephone with the surgeon’s office, though some were not informed and were simply still waiting for a date for surgery:


*“His office called and had said that they’re off… At that time, they were just closing the ORs for, they were hoping, for a 2-month period. And then they had given me a new date, and had said that, you know, they’re hoping to be open by this time, he should be able to get in.”*
[Participant 2].


*“I just didn’t hear anything until it was time.”*
[Participant 16].

Many participants did relay how they were pleased with the quality of their interactions with their child’s surgical team during the pandemic.


*“They were understanding… there’s not a lot that they have control over, I guess. …they can’t always just make something happen tomorrow, type of thing. But she was very good to like check-up and passed messages along and reassured me that she would try to get her in as soon as possible…”*
[Participant 18].

### 3.4. Levels of Distress

Despite coping well during the pandemic and not experiencing additional distress as it related to health-associated fears, participants reported varying levels of frustration and distress due to their child’s delay in surgery. Many reported uncertainty, disappointment, and frustration with the surgical delay, though they understood the reasons for the delays. Several participants also discussed how the delay was experienced by their child. Two participants described these feelings as:


*“The second one, when they changed it again, it was a little bit more frustrating. Because like I said, he’s had two more ear infections and things like that. And my son has a little bit of anxiety. So, waiting for it, on top of everything else, it’s been causing him a lot of anxiety. So, I was a little bit more frustrated...”*
[Participant 2].


*“The knowledge of the surgery is going to happen, but not knowing when it’s going to happen is difficult for everybody… it’s difficult for a little guy. You know, he’s nervous about it. So that would be it—prolonging that period where he knows about it.”*
[Participant 11].

Participants also indicated their physical and mental health had been impacted by their emotional distress and worry related to both COVID-19 and associated delays in treatment for their child.


*“You have this feeling in your body. Like it’s just constant anxiety of waiting. … It’s excruciating. Especially when it’s your child. Because you’re looking at him and he’s turning a little bit more blue, and you’re like should we take him in?”*
[Participant 10].


*“I got a hold of my family doctor and I told him how overwhelmed I was, and he prescribed me… kind of like an antidepressant.”*
[Participant 27].

Participants also feared that the COVID-19 surgical delay would result in disease progression, negative consequences, or developmental and social delays for their child. As described by two participants:


*“Well, the most difficult thing overall was just the worry… could she be losing vision permanently while we’re waiting?”*
[Participant 18].


*“I guess the big concern is… he’s continuing to grow which means his heart is continuing to grow. … If it’s a really long delay, I get the impression that that could make the actual surgery more difficult… Concern #2… I would really love him to be able to start primary with his twin brother and be there for the start of the year.”*
[Participant 11].

Many participants communicated they were relieved that their child was not waiting for a “lifesaving” surgery. In this way, they minimized the severity of the distress they experienced. One participant put it this way:


*“I mean we’re very lucky that his procedure wasn’t like as life threatening as some of the others were waiting for a surgery date as well.”*
[Participant 5].

### 3.5. Coping with COVID-19

For the most part, participants did not differentiate coping with the pandemic from coping with prolonged surgical delays experienced by their child. Participants therefore described coping with these issues in similar ways. While participants’ daily routines had changed due to region wide COVID-19 restrictions, they discussed coping well during the pandemic. Although several participants reported some financial and employment challenges due to COVID-19, for most, pandemic-related restrictions themselves did not directly compound their stress related to their child’s surgical delay. As stated by one participant:


*“… with my life, it actually changed myself working in an office to a virtual environment. So now my company went 100% virtual. So being home, I can work home and I’m not going in and out of the office. Which is a nice thing right now, especially with being pregnant and having a child with special needs. It helps with your work-life balance.”*
[Participant 12].

Participants reported activities such as being physically active, reading, watching television, and working as ways to cope with the pandemic-related restrictions. As described by one of the participating parents.


*“I worked more. That would probably have been a big support. Where, like I think society struggled a lot, like not being able to socialize or to have like human interaction. But I worked more. So that helped a lot to actually talk and see people.”*
[Participant 5].

All participants discussed strong social support systems through close family and friends and how these support systems reduced their distress. For instance, one participant said:


*“Well, just we have like a good family. … I haven’t really seen anybody. We never really bubbled with anybody. But I have a supportive… Like my husband’s supportive, and our parents are supportive. And we talk to everybody, and we Facetime.”*
[Participant 1].

Most participants reported low levels of COVID-19 related worry (i.e., being infected by SARS-CoV-2), with many discussing the comparatively low cases in the region and behaviour modifications as reasons for their limited concern.


*“I’m doing my part. I’m staying home, I’m washing my hands, I’m being very cautious. And if I have to go out, just like we try our best not to go out every day. Obviously shopping, it’s essential. We would just literally go and pack up for like a week, and then if need be, we’d go. But we were very staying home, following the guidelines of the Health [Authority] of Nova Scotia.”*
[Participant 12].

### 3.6. Hospital Restrictions

Most participants experienced the COVID-19-related hospital restrictions as difficult and oftentimes distressing once their child entered the hospital for surgery. This related predominantly to institutional restrictions limiting in-hospital caregivers to one caregiver only. Thus, participants were often the sole parent in-hospital with their child. Participants expressed distress over not having another support person there primarily for their own emotional needs, not necessarily as an additional support for their child.


*“The only thing that was a little challenging, I find, especially with me because I’m… I was at the time three and a half months pregnant, and I’m alone at the hospital. I couldn’t have my husband with me up in like the recovery unit. It was just anxious—‘Oh my God, what’s going on?…’”*
[Participant 12].

Similarly, several participants discussed the distress of waiting at home on the surgery day because the other parent was the child’s support person at the hospital:


*“…Like the day surgery was horrible… Like we’re on the phone, and I said this is just the worst feeling not being there. And not knowing, like I didn’t know how scary it had gotten in the OR until days later. … Like you’re listening to them but there’s teams of people talking at you. And when there was two of us [there], one of us would always be comprehensive. Like we would be able to understand and we would kind of talk to the other one. But this is… It was really hard. … I understand the precautions but I don’t agree with it whatsoever, especially when it’s something like this. Because if something had gone the other way, I would have been vicious.”*
[Participant 10].

At the same time, upon entering the hospital, some people were comforted by the precautions in place to prevent the spread of COVID-19, with a high level of trust in the surgical team and the hospital in general.


*“Once you’re there, you feel the procedure is in place. So it was, you know, everybody had masks. There was, you know… You could tell that there were policies and checks to keep people safe.”*
[Participant 13].

## 4. Discussion

In this study of parents with children facing prolonged wait times for elective, non-emergent surgery during the first wave of the COVID-19 pandemic, participants were found to have substantial levels of psychosocial distress. Parents did not necessarily feel additional distress as it related to the transmission of SARS-CoV-2, but instead experienced distress as it related to prolonged wait times and hospital restrictions. Parents feared progression of their child’s disease and the possible negative repercussions for social and educational development.

In a large study assessing parental experiences in pediatric surgery, Miller and colleagues found that 95% of parents felt waiting for their child’s surgery caused emotional distress for their family [1]. Furthermore, they determined that despite the procedures in question being considered elective, non-urgent surgeries, half of parents felt their child’s health was deteriorating, and over three quarters of parents anticipated that undergoing surgery would substantially improve their child’s quality of life. Unfortunately, in a recent editorial, the Pediatric Surgical Chiefs of Canada have outlined significant challenges during the COVID-19 pandemic [4]. Over 7600 pediatric surgeries were postponed across Canada by June of 2020, with an additional 4000 children not having access to initial surgical consultations. In the current study, prolonged wait time and hospital restrictions have served to further increase the anxiety and distress experienced by parents.

In our study, two additional factors linked with increased psychosocial distress for parents were hospital restrictions and quality of communication with the healthcare team. Parents experienced distress due to inability to have a second support person with them during the perioperative period. Additionally, poor communication with the healthcare team lead to higher levels of distress as described by participants. Our institution has since expanded support allowance, whereby a second adult support person is able to be in hospital for select cases. Furthermore, our institution has begun publicly posting COVID-19-related updates online for parents to view. Additional system level changes have been implemented in the literature, including centralized referral and triage systems [15].

The greatest burden of distress may have been experienced by parents of children who were required to travel from out of province and thus were required to self-isolate, had experienced multiple delays, and who had what was perceived as more severe disease with a higher risk of developmental delay. Inherent to complex pediatric surgical care, this unique group is relatively common and represents an important cohort for targeted support in the future. Interestingly, parents who possessed a higher degree of familiarity with their child’s diagnosis and pending procedure expressed lower levels of psychosocial distress during interviews. Improved parent education during initial pre-operative consultations may be an avenue for such targeted supports in reducing distress.

This study has limitations that must be considered. This study takes place in a universal, single-payer healthcare system and thus our findings may not be transferable to other health systems. Further, Canada has been relatively well positioned amongst other countries in terms of the severity of COVID-19, and the province in which this study took place has lower COVID-19 cases per capita than some other jurisdictions in Canada. However, we hypothesize that a greater COVID-19 burden would only serve to worsen both surgical delays and parental distress. Transferability of our findings may be affected by the educational levels and the proportion of married or common-law participants in this study, both of which were high. Similarly, all participants in this study identified as female; male caregivers may express different feelings towards COVID-19 and prolonged surgical wait times. No predetermined theoretical framework was utilized in this study. Identification of applicable frameworks, or development of an entirely new framework as it relates to psychosocial distress, may be an interesting avenue for future research. Lastly, 12 participants ultimately did not participate in the semi-structured interview and may represent unique experiences which were not captured. However, only two of these participants actively expressed a desire to forego an interview, while the remainder were not reachable by telephone and thus may have similar viewpoints to those who ultimately took part in the study.

This is the first study to investigate the psychosocial distress experienced by parents with children facing prolonged wait times as a result of the COVID-19 pandemic. The pandemic continues to present challenges across the globe. Previous models have suggested clearing surgical backlogs may take many months, if not years [16]. As the pandemic timeframe extends, this backlog may continue to grow. Should the pandemic finally be controlled, previous studies have, regardless, suggested poor performance on meeting wait list targets at baseline. Surgical wait time targets for pediatric surgery in Canada are met in only 65% of cases and thus the findings in this study may be useful in creating targeted psychosocial interventions outside of the pandemic [3]. Additional research must be focused on creating these interventions and implementing them to the benefit of our patient populations. For example, previous studies have reported that parents desire psychosocial interventions that support social engagement, facilitate communication with providers, and promote self-care [17].

## 5. Conclusions

In this cross-sectional, qualitative analysis of parents with children experiencing prolonged surgical wait times during the COVID-19 pandemic, participants expressed increased levels of psychosocial distress. The findings of this study support further research on the development and implementation targeted interventions to improve both parent and child experiences during the pre-operative and peri-operative phases of care.

## Figures and Tables

**Table 1 children-09-00087-t001:** Participant characteristics.

Variable	*N*
Total participants	18
*Demographics*	
Sex of parent	
Female	18
Male	0
Sex of child	
Female	5
Male	13
Education (highest level of completion)	
Less than high school	0
High school	3
College or undergraduate university	13
Postgraduate university or professional program	2
Living situation	
Sibling of child in home	12
Parent’s partner in home	13
Others in home (roommates, other family members)	1
Other (people other than those above)	0
None of the above	1
Marital status	
Married	12
Single/never married	4
Divorced/separated	1
Common-law	1
*Clinical Characteristics*	
Number of child’s medical history diagnoses	
0	0
1	7
2	0
3+	1
Prior surgical history/experiences for child	
Yes	9
No	9
Parent’s past surgical experiences	
Yes	1
No	17
Surgical service	
Ophthalmology	1
Otolaryngology	5
Cardiac	2
General surgery	1
Urology	2
Plastic	2
Orthopaedics	4
Neurosurgy	1
Date for surgery received prior to interview	
Yes	13
No	5
Surgery completed prior to interview	
Yes	10
No	8

## Data Availability

The data presented in this study are available on request from the corresponding author. The data are not publicly available in order to protect privacy and confidentiality of the research participants.

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
