# Peer review of "Psychosocial Distress in Parents with Children Awaiting Surgery during the COVID-19 Pandemic"

_children, 2022, doi:10.3390/children9010087_

Round 1

Reviewer 1 Report

Even when psychosocial stress has been widely studied in the psychological field, is uncommon to find it in medical research. That is what makes this research commendable, taking into account the difficulties to organize and launch it during the first phase of the pandemic. 

Author Response

Thank you for your consideration of our manuscript entitled Psychosocial distress in parents with children awaiting surgery during the COVID-19 pandemic. We have addressed each of the reviewer comments below, point by point. We have provided a tracked copy of the revised manuscript.

Reviewer 1:

Even when psychosocial stress has been widely studied in the psychological field, is uncommon to find it in medical research. That is what makes this research commendable, taking into account the difficulties to organize and launch it during the first phase of the pandemic.

Thank you for taking the time to review our paper. We appreciate the kind words and recognition, and hope this study can guide future care for patients and caregivers.

Reviewer 2 Report

The qualitative study intends to trace the relationship between the profile of the parent's psychosocial condition and the child's postponement of the surgery during the COVID 19. While it is true that any contribution relating to the study of social phenomena induced by the pandemic should be accepted with interest, it is true, however, that the interest in this study cannot be expressed with its publication as a scientific article but only as a case report.

This is a study that has serious methodological gaps, some of which have been declared by authors:

 - There is no a control group and methodological procedures are not described, perhaps because they are not defined, as the design is insufficient on this level.

- We do not see the theoretical model used both in the reading of psychosocial distress and in the emotional resonance that are indicated in the responses to the interview.

- There is no theoretical and structural analysis (categories, constructs, etc.) in the presentation of the interview.

While providing a detailed analysis relating to emotional states, there is no attempt to give a methodological organicity to the qualitative data.

The work must therefore be transformed into a case report.

The qualitative study intends to trace the relationship between the profile of the parent's psychosocial condition and the child's postponement of the surgery during the COVID 19. While it is true that any contribution relating to the study of social phenomena induced by the pandemic should be accepted with interest, it is true, however, that the interest in this study cannot be expressed with its publication as a scientific article but only as a case report.

This is a study that has serious methodological gaps, some of which have been declared by authors:

 - There is no a control group and methodological procedures are not described, perhaps because they are not defined, as the design is insufficient on this level.

- We do not see the theoretical model used both in the reading of psychosocial distress and in the emotional resonance that are indicated in the responses to the interview.

- There is no theoretical and structural analysis (categories, constructs, etc.) in the presentation of the interview.

While providing a detailed analysis relating to emotional states, there is no attempt to give a methodological organicity to the qualitative data.

The work must therefore be transformed into a case report.

Author Response

Thank you for your consideration of our manuscript entitled Psychosocial distress in parents with children awaiting surgery during the COVID-19 pandemic. We have addressed each of the reviewer comments below, point by point. We have provided a tracked copy of the revised manuscript.

Reviewer 2:

The qualitative study intends to trace the relationship between the profile of the parent's psychosocial condition and the child's postponement of the surgery during the COVID 19. While it is true that any contribution relating to the study of social phenomena induced by the pandemic should be accepted with interest, it is true, however, that the interest in this study cannot be expressed with its publication as a scientific article but only as a case report.

Thank you for taking the time to review our paper. We agree that scientific rigor is an important requirement for all peer-reviewed publications. However, we respectfully disagree that our study should be presented as a “case report.” As described in our paper, this was a qualitative study, underpinned by an interpretivist paradigm and using a well-established data analysis approach (thematic analysis). Thus, this is an empirical research study using robust qualitative methods. Thank you again for reviewing our paper – it is greatly appreciated.

This is a study that has serious methodological gaps, some of which have been declared by authors:

 - There is no a control group and methodological procedures are not described, perhaps because they are not defined, as the design is insufficient on this level.

This study was designed with input from multiple clinicians and experts in qualitative research. Further, the paper is reported in accordance with the Standards for Reporting Qualitative Research (SRQR) guidelines, as outlined in Lines 64-65, in order to improve transparency of its conduction and results.

To reiterate, this study was a cross-sectional qualitative study (Line 62), using purposive sampling (Line 74) until theory saturation (Line 75), whereby semi-structured interviews (Line 87 and Supplemental Material) were utilized for data generation. Qualitative analysis was achieved through an interpretivism research paradigm (Line 91) with thematic analysis under a Braune and Clarke Framework (Line 93). The first three interviews were individually coded by two researchers in order to develop a codebook that was applied and updated to sequential interviews (Lines 94-95). Consensus discussion was performed by multiple authors after full analysis by the two researchers previously outlined (Lines 96-97)

We acknowledge in our limitations that there is no comparator group by which to judge the experiences of the participants (Lines 267-270). However, unlike in quantitative research such as cohort or case-control studies, where a control group is necessary to estimate an effect size of an intervention, qualitative research has no such necessity. In fact, it would be unusual to use a control group in qualitative research wherein the aim of qualitative inquiry is to gather rich data about individuals’ experiences around some phenomenon. Individuals who have not experienced that phenomenon would not be of interest.

- We do not see the theoretical model used both in the reading of psychosocial distress and in the emotional resonance that are indicated in the responses to the interview

As this study was performed with an interpretivism research paradigm and thematic analysis, no conceptual model of psychosocial distress was defined a priori and all coding was performed in an inductive manner. The semi-structured interview questions were open ended and did not specifically include the use of “psychosocial distress” and instead allowed participants to define their own experiences, in keeping with an interpretivism research paradigm whereby participant experiences are interpreted in the context of their environment and lived experiences.

We have provided additional description of the inductive coding used within the methods section of the paper:

Lines 94-95, “The first three interviews were independently coded (PKL, RU) and used to develop a novel codebook using inductive coding performed in an iterative fashion.

- There is no theoretical and structural analysis (categories, constructs, etc.) in the presentation of the interview.

The objective of this study was not to define new constructs or theoretical models of psychosocial distress. Such a study would require a markedly different approach and could be an avenue for future research. Instead, we sought to explore psychosocial outcomes of parents with children facing prolonged surgical wait times during the COVID-19 pandemic and describe the themes within those experiences (Lines 57-59).

We present four identified themes (Lines 113-117), including communication, levels of distress, coping with COVID-19, and hospital restrictions. Descriptions of these themes, as well as supporting quotations are provided. As discussed in a response to Reviewer #3, we also provide comparison examples. We also provide supporting codes and domains which informed the construction of our themes (Supplement Table S3). 

While providing a detailed analysis relating to emotional states, there is no attempt to give a methodological organicity to the qualitative data.

The work must therefore be transformed into a case report.

We hope we have clarified the methodological rigor by which this qualitative research study was developed and performed.

Reviewer 3 Report

This manuscript is meaningful in that it deals with psychosocial distress in parents with children awaiting surgery especially during the COVID-19 pandemic. In addition, the result guides the direction of the further research in the relevant field as the authors clearly mentioned that “the findings of this study support further research on the development and implementation targeted interventions to improve both parent and child experiences during the pre-operative and peri-operative phases of care”. However, the authors did not mention how to ensure the trustworthiness (reliability and validity) of their qualitative study. I strongly insist that this issue should be clearly mentioned in the method section.

Here are a few comments to consider.

Page 3; Within (  ), what is PKL, RU, DF, PH? I found out that those are the names of authors who did the work only after I finished reading. It should be explained beforehand if the authors insist…But, I don’t think it’s not necessary….

I guess it would be helpful to summarize the themes of semi-structure interview questions in the manuscript for readers to understand the findings better.

Thank you.

Author Response

Thank you for your consideration of our manuscript entitled Psychosocial distress in parents with children awaiting surgery during the COVID-19 pandemic. We have addressed each of the reviewer comments below, point by point. We have provided a tracked copy of the revised manuscript.

Reviewer 3:

This manuscript is meaningful in that it deals with psychosocial distress in parents with children awaiting surgery especially during the COVID-19 pandemic. In addition, the result guides the direction of the further research in the relevant field as the authors clearly mentioned that “the findings of this study support further research on the development and implementation targeted interventions to improve both parent and child experiences during the pre-operative and peri-operative phases of care”. However, the authors did not mention how to ensure the trustworthiness (reliability and validity) of their qualitative study. I strongly insist that this issue should be clearly mentioned in the method section.

Thank you for taking the time to review our paper – we appreciate the effort in doing so. We agree on the importance of this study in both describing current distress as well as identifying areas of future research.

The concepts of reliability and validity in qualitative research differ from that of quantitative research and there is argument as to whether the terms even apply to qualitative research (Noble and Smith 2015). Alternate definitions of reliability and validity have been proposed by some authors, including truth value (or credibility) in place of validity, and consistency and neutrality (or dependability) in place of reliability (Lincoln and Guba 1985).

When considering “truth value” in place of validity, which is typically thought of as the precision by which the findings accurately reflect the data in quantitative research, authors and readers must acknowledge that multiple realities may exist when interpreting qualitative data. As such, the research must define their personal experiences and viewpoints and acknowledge that interpretations may be affected by these experiences. By providing Reflexivity Statements for the four authors who performed thematic analysis, we have provided for this truth value (Supplemental Figure S2).

“Consistency” and “neutrality” aim to mirror reliability in quantitative research, where authors account for personal and research biases to provide consistency in the analytical procedures. We have highlighted all main steps in our research protocol, including which authors performed interviews, coding, analysis, etc. This includes a detailed description of the codes developed, the subsequent domains applicable to those codes, and lastly the themes common amongst those domains (Supplemental Table S3 and Results).

We have highlighted the validity (ie, credibility) and reliability (ie, dependability) within the methods section more clearly:

Lines 93-101, “In order to promote credibility and dependability of our analysis13, the first three interviews were independently coded (PKL, RU) and used to develop a novel codebook using inductive coding performed in an iterative fashion. The remaining interviews were coded to develop domains (PKL, RU). Themes summarizing these domains were developed (PKL, RU). Consensus discussion and elaboration on domains and themes was performed with multiple team members (DF, PKL, RU, PH) to support reliability in our procedures13.

Noble, H., & Smith, J. (2015). Issues of validity and reliability in qualitative research. Evidence-based nursing, 18(2), 34-35.

Lincoln YS, Guba EG. Naturalistic inquiry. Beverly Hills, CA: Sage, 1985.

Here are a few comments to consider.

Page 3; Within ( ), what is PKL, RU, DF, PH? I found out that those are the names of authors who did the work only after I finished reading. It should be explained beforehand if the authors insist…But, I don’t think it’s not necessary….

We believe providing the author initials are an important detail that allows linkage of the methods to the Reflexivity Statements which, as outlined in above responses to Reviewer #2 and Reviewer #3, are essential components in providing transparency within qualitative researcher.

As initials of an author are provided first on line 84 “…subsequently by a members of the research team (PKL)” we believe the notation is introduced appropriately and have not made additional amendments.

I guess it would be helpful to summarize the themes of semi-structure interview questions in the manuscript for readers to understand the findings better.

The semi-structured interview guide is provided in full in the Supplemental Material. We have provided an additional summary in the methods section for ease of the reader.

Lines 88-90, “The semi-structured interview guide sought to elicit participant experiences with the pandemic, surgical delays, and their impact on the participant and their children.

Round 2

Reviewer 2 Report

The additions included in the text have certainly improved the contribution.  Procedures indicated in the response to the reviewer, who declared their absence, had to be indicated already in the first draft.  As regards the possibility of a control group, a comparison between 2 groups could be hypothesized, considering for example another clinical group: psychosocial stress in a group of parents of children for whom surgery was delayed and a group of parents whose children have delayed the follow up due to a disease

I’m absolutely disagree with the choice not to assume a theoretical model for reading the social phenomenon (taken from the reference literature, from studies based on scientific evidence). The interpretative paradigm does not justify the absence of a model, on the contrary it makes it necessary, making the interpretation less arbitrary.

The suggestion of a case report would have allowed this denial by taking an exploratory approach.

Finally authors said too less about interview.

Author Response

David Forner MD MSc

Division of Otolaryngology – Head & Neck Surgery

Dalhousie University

Paul Hong MD MSc

Professor

Division of Otolaryngology – Head & Neck Surgery

Dalhousie University

Attention:

Sari A. Acra MD MPH

Editor-in-Chief, Children

Re: children-1450645

Dear Dr. Acra,

Thank you for your consideration of our manuscript entitled Psychosocial distress in parents with children awaiting surgery during the COVID-19 pandemic. We have addressed the reviewer comments below, point by point. We have provided a tracked copy of the revised manuscript.

Reviewer #2:

The additions included in the text have certainly improved the contribution.  Procedures indicated in the response to the reviewer, who declared their absence, had to be indicated already in the first draft.  As regards the possibility of a control group, a comparison between 2 groups could be hypothesized, considering for example another clinical group: psychosocial stress in a group of parents of children for whom surgery was delayed and a group of parents whose children have delayed the follow up due to a disease.

Thank you for again taking the time to review our manuscript. We appreciate your time in doing so, and for helping improve our manuscript.

We agree that an exploration of another group of parents, whom which there was no delay or an alternative reason for delay, may be an interesting avenue for future inquiry. We make note of this already in our limitations (lines 270-273). However, such alternative groups would have had, predictably, different experiences and would not have the lived experiences of the group we were interested in – those whom which had surgical delays due to COVID-19. As stated in a previous response, these alternative situations were not of interest, as our goal was to gather rich data about individuals’ experiences around the phenomena of delay due to COVID-19.

We have now included this specifically in the limitations, which now reads:

Lines 270-276, “While participants were asked to describe their experiences as they relate specifically to prolonged surgical wait times due to the COVID-19 pandemic, there is no comparator group of parents with children who either experienced no delay during the COVID-19 pandemic or experienced delay for non-pandemic related reasons. However, the goal of qualitative inquiry is to gather rich data about individuals’ experiences around some phenomenon. Individuals who have not experienced that phenomenon, in this case prolonged surgical wait times due to COVID-19 related restrictions, would not be of interest.”

I’m absolutely disagree with the choice not to assume a theoretical model for reading the social phenomenon (taken from the reference literature, from studies based on scientific evidence). The interpretative paradigm does not justify the absence of a model, on the contrary it makes it necessary, making the interpretation less arbitrary.

We have now included this as a limitation and area of future work.

Lines 286-288, “No predetermined theoretical framework was utilized in this study. Identification of applicable frameworks, or development of an entirely new framework as it relates to psychosocial distress, may be an interesting avenue for future research.”

The suggestion of a case report would have allowed this denial by taking an exploratory approach.

A case report refers to a single case with detailed descriptions of the clinical presentation, comorbidities, disease diagnosis and treatment, and follow-up of outcomes and complications. In qualitative research, a case report would generally not be helpful as a single patient cannot generate the rich, representative data needed to explore an entire group’s experiences. On the other hand, case studies have been used in healthcare research to generate in-depth understanding of issues. However, case studies employ qualitative methodology and there is little established distinction between a case study and other qualitative research studies. In fact, we argue that case studies may also employ theoretical frameworks, or not, and thus this distinction is not helpful for our study (Crowe 2011).

Crowe, S., Cresswell, K., Robertson, A., Huby, G., Avery, A., & Sheikh, A. (2011). The case study approach. BMC medical research methodology11(1), 1-9.

Finally authors said too less about interview.

The entire, word-for-word, semi-structured interview is available to the reader in the supplemental material. We have included two additional sentences, and believe the reader’s attention can be turned to the supplemental material to gain a full understanding of exactly what was included in the interview guide.

Lines 90-92, “Open ended questions were utilized. For example, we asked “Tell me about your experiences since your child’s surgery was delayed because of the COVID-19 pandemic.”

Thank you again for considering this paper for publication in Children.

Sincerely,

David Forner and Paul Hong,
